# Towards Learning to Reason: Comparing LLMs with Neuro-Symbolic on Arithmetic Relations in Abstract Reasoning

**Michael Hersche[1], Giacomo Camposampiero[1,2], Roger Wattenhofer[2], Abu Sebastian[1], Abbas Rahimi[1]**

[1]IBM Research - Zurich
[2]ETH Zürich

## Abstract

This work compares large language models (LLMs) and neuro-symbolic approaches in solving Raven's progressive matrices (RPM), a visual abstract reasoning test that involves the understanding of mathematical rules such as progression or arithmetic addition. Providing the visual attributes directly as textual prompts, which assumes an oracle visual perception module, allows us to measure the model's abstract reasoning capability in isolation. Despite providing such compositionally structured representations from the oracle visual perception and advanced prompting techniques, both GPT-4 and Llama-3 70B cannot achieve perfect accuracy on the `center` constellation of the I-RAVEN dataset. Our analysis reveals that the root cause lies in the LLM's weakness in understanding and executing arithmetic rules. As a potential remedy, we analyze the Abductive Rule Learner with Context-awareness (ARLC), a neuro-symbolic approach that *learns to reason* with vector-symbolic architectures (VSAs). Here, concepts are represented with distributed vectors s.t. dot products between encoded vectors define a similarity kernel, and simple element-wise operations on the vectors perform addition/subtraction on the encoded values. We find that ARLC achieves almost perfect accuracy on the `center` constellation of I-RAVEN, demonstrating a high fidelity in arithmetic rules. To stress the length generalization capabilities of the models, we extend the RPM tests to larger matrices ($3 \times 10$ instead of typical $3 \times 3$) and larger dynamic ranges of the attribute values (from 10 up to 1000). We find that the LLM's accuracy of solving arithmetic rules drops to sub-10%, especially as the dynamic range expands, while ARLC can maintain a high accuracy due to emulating symbolic computations on top of properly distributed representations.[1]

## 1  Introduction

Abstract reasoning is often regarded as a core feature of human intelligence. This cognitive process involves abstracting rules from observed patterns in a source domain, and applying them in an unseen target domain. With the ultimate aim to achieve human-level intelligence, abstract reasoning tasks have sparked the interest of many in machine learning research. Thanks to the availability of large datasets (Barrett et al. 2018; Zhang et al. 2019; Hu et al. 2021), vari-

ous learning-based methods, ranging from pure connectionist (Benny, Pekar, and Wolf 2021; Wu et al. 2020) to neuro-symbolic (Zhang et al. 2021, 2022; Hersche et al. 2023, 2024a; Camposampiero et al. 2024; Sun et al. 2025) approaches, achieved promising results in this domain.

More recently, the zero- and few-shot capabilities of LLMs and their multi-modal variants have been tested on various abstract reasoning tasks such as verbal (Webb, Holyoak, and Lu 2023; Stevenson et al. 2023; Gendron et al. 2024; Lewis and Mitchell 2024) or visual (Cao et al. 2024; Webb, Holyoak, and Lu 2023; Hu et al. 2023; Mitchell, Palmarini, and Moskvichev 2024; Camposampiero et al. 2023; Jiang et al. 2024; Ahrabian et al. 2024; Zhang et al. 2024; Wüst et al. 2024; Latif et al. 2024; Lewis and Mitchell 2024) analogies. One natural approach towards zero-shot visual abstract reasoning is to leverage multi-modal LLM's vision capabilities to solve the task end-to-end. However, these multi-modal models perform significantly worse than their text-only version (Mitchell, Palmarini, and Moskvichev 2024), which might stem from a missing fine-grained compositional feature comprehension (Cao et al. 2024). As an additional help, LLMs have been provided with text-only inputs by giving them access to an oracle perception, i.e., providing perfectly disentangled representations (Webb, Holyoak, and Lu 2023; Hu et al. 2023). While this improves their reasoning abilities, LLMs still fail to achieve perfect accuracy on many simple tasks. One example is represented by Raven's progressive matrices (RPMs) (Raven, Court, and Raven 1938), a benchmark that tests visual abstract reasoning capabilities by measuring the fluid intelligence of humans. Here, the state-of-the-art (SOTA) LLM-based approach (Hu et al. 2023) achieves only 86.4% accuracy in the `center` constellation of I-RAVEN (Hu et al. 2021), which we observe to be a gate-keeper for this task (see Section 2).

In contrast, recent neuro-symbolic approaches showed not only almost perfect accuracy on the `center` constellation of I-RAVEN, but also demonstrated high fidelity in out-of-distribution (OOD) settings. For instance, the Abductive Rule Learner with Context-awareness (ARLC) represents attribute values with high-dimensional, distributed representations based on vector-symbolic architectures (VSAs) (Plate 1995, 2003; Gayler 2003; Kanerva 2009). This allows to perform probabilistic abductive reasoning in superposition, notably reducing the compute and memory demand that is usually dominated by weighted model count-

[1]Our code is available at https://github.com/IBM/raven-large-language-models.

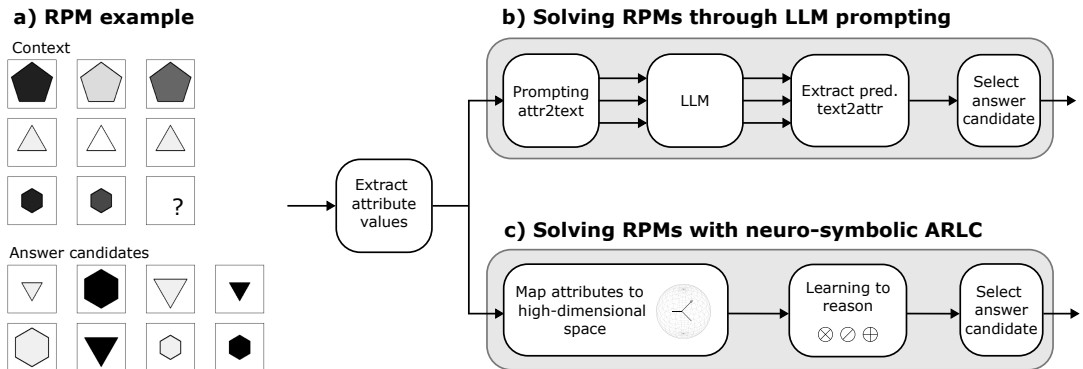

Figure 1: This work compares the abstract reasoning capabilities of large language models (LLMs) and neuro-symbolic ARLC on Raven's progressive matrices (RPM) tests. **a)** An RPM example taken from the `center` constellation of I-RAVEN. The task is to find the empty panel at the bottom-right of the context matrix by selecting one of the answer candidates. **b)** Solving RPMs through LLM prompting. Visual attribute values are extracted from the I-RAVEN dataset and assembled to individual per-attribute text-only prompts. LLMs are prompted to predict the attribute of the empty panel. Finally, the attribute predictions are compared with the answer candidates, whereby the best-matching answer is selected as the final answer. **c)** Solving RPMs with neuro-symbolic ARLC that relies on distributed similarity-preserving representations and manipulates them via dimensionality-preserving operations; it learns rule-formulations as a differentiable assignment problem.

ing (Zhang et al. 2021; Raedt et al. 2015). Learning the RPM rules becomes a differentiable assignment problem of high-dimensional panel representations in a series of binding and unbinding operations, which can be solved with unconstrained optimization algorithms such as stochastic gradient descent (SGD). ARLC outperformed the SOTA LLM-based approach (Hu et al. 2023) both on in-distribution and OOD, thanks to relying on structured and similarity-preserving representations based on fractional power encoding (FPE) (Plate 2003). Moreover, ARLC's rules could still be manually programmed and further trained allowing to extend the knowledge of the model, rather than completely erasing it as shown in other settings (Wu, Zhang, and Shu 2019).

This paper extends on the initial work on ARLC (Camposampiero et al. 2024), by comparing its abstract reasoning capability with two prominent LLMs, GPT-4 (OpenAI et al. 2024) and Llama-3 70B (Dubey et al. 2024) (see Figure 1). Circumventing the perception by providing ground-truth attribute labels to the models allows us to measure their analogical and mathematical reasoning capabilities in isolation when such *compositionally structured* (i.e., disentangled) representations are provided. Our comprehensive prompting efforts lead to very high accuracy for Llama-3 70B (85.0%) and GPT-4 (93.2%), where the latter notably outperforms previous reports with GPT-3 (Hu et al. 2023) (86.4%) and GPT-4 o1-preview (Latif et al. 2024) (18.00%). The LLM's still imperfect accuracy on the isolated task motivated us to further analyze their capability of detecting and executing different rules. In both GPT-4 and Llama-3 70B, we find a notable weakness in performing arithmetic rules that require row-wise additions or subtractions (e.g., see the last prompt in Figure 2). To gain more insight about this behavior, we set up a new RPM dataset (I-RAVEN-X) that increases the grid size from 3×3 to 3×10, additionally allowing for a configurable dynamic range for the arithmetic computations. Also

here, we observe a notable weakness in the arithmetic rule that gets even amplified by an increasing dynamic range. On the other hand, ARLC demonstrates high accuracy on larger grid sizes and allows to increase the dynamic range without requiring further retraining, thanks to the the capability of adjusting the underlying structured FPE representations.

## 2  Datasets

### I-RAVEN

We test the models on the `center` constellation of I-RAVEN (Hu et al. 2021) (see Figure 1). The test consists of a 3×3 context matrix and eight answer candidate panels. Each panel contains an object, characterized by different attributes (shape, size, and color). The relation between each attribute's value in different panels is governed by a well-defined set of rules: `constant`, `progression`, `arithmetic`, and `distribute three`. The task is to infer the rule governing each attribute in the context matrix and use it to determine the content of the missing (bottom-right) panel, selecting it within the eight candidate answers. Compared to other RPM benchmarks that have been used to evaluate LLMs (Webb, Holyoak, and Lu 2023), I-RAVEN tests a more complex range of logical and arithmetic skills. While I-RAVEN provides tests in various constellations with more objects that may intuitively appear more arduous to solve, LLMs are more challenged with the seemingly simple constellations. For instance, GPT-3 achieved a higher accuracy on the `2x2` and `3x3` constellations (78.0% and 86.4%) than on `center` (77.2%) (Hu et al. 2023). Moreover, high accuracy can be maintained on the `2x2` and `3x3` constellations while only looking at the last row of the context matrix (Hu et al. 2023), effectively showing that no analogical reasoning is required to solve the test in these constellations. Hence, we opted to focus our evaluation on the `center` constellation only, using 500 samples from I-RAVEN's test

**a) LLM prompts for I-RAVEN**

System: Complete the Raven's progressive matrix:
User: Only return the missing numbers!
  row 1: 5, 5, 5;
  row 2: 3, 3, 3;
  row 3: 6, 6,
Output: 6

Attribute: **shape**
Rule: **constant**
Correct answer: **6**

System: Complete the Raven's progressive matrix:
User: Only return the missing numbers!
  row 1: 6, 6, 6;
  row 2: 4, 4, 4;
  row 3: 2, 2,
Output: 2

Attribute: **size**
Rule: **constant**
Correct answer: **2**

System: Complete the Raven's progressive matrix:
User: Only return the missing numbers!
  row 1: 8, 2, 6;
  row 2: 1, 0, 1;
  row 3: 8, 7,
Output: 6

Attribute: **color**
Rule: **arithmetic -**
Correct answer: **1**

**b) LLM prompts for our new I-RAVEN-X**

System: Complete the Raven's progressive matrix:
User: Only return the missing numbers!
  row 1: 320, 322, 324, 326, 328, 330, 332, 334, 336, 338;
  row 2: 718, 720, 722, 724, 726, 728, 730, 732, 734, 736;
  row 3: 224, 226, 228, 230, 232, 234, 236, 238, 240,
Output: 242

Attribute: **shape**
Rule: **progression**
Correct answer: **242**

System: Complete the Raven's progressive matrix:
User: Only return the missing numbers!
  row 1: 73, 73, 73, 73, 73, 73, 73, 73, 73, 73;
  row 2: 677, 677, 677, 677, 677, 677, 677, 677, 677, 677;
  row 3: 695, 695, 695, 695, 695, 695, 695, 695, 695,
Output: 695

Attribute: **size**
Rule: **constant**
Correct answer: **695**

System: Complete the Raven's progressive matrix:
User: Only return the missing numbers!
  row 1: 769, 667, 0, 4, 2,   20, 63,  3,   5,   5;
  row 2: 848, 0,   0, 0, 387, 2,  106, 7,   308, 38;
  row 3: 611, 2,   0, 0, 0,   0,  0,   551, 0,
Output: 352

Attribute: **color**
Rule: **arithmetic -**
Correct answer: **58**

Figure 2: Prompting LLMs to predict the missing content (value) of the RPM. Correct ouputs are marked green and wrong ones red. **a)** Individual per-attribute text-only prompts to solve RPM tasks from I-RAVEN. **b)** Example prompts with of our novel configurable I-RAVEN-X dataset of size $3 \times 10$ with a value range of $m = 1000$. In both the I-RAVEN and I-RAVEN-X examples, the LLM (GPT-4) errs in the arithmetic rules.

set. Inspired by recent works (Webb, Holyoak, and Lu 2023; Hu et al. 2023), we simplify RPM from a visual abstract reasoning test to a purely abstract reasoning test. Assuming a perfect perception, we extract the attribute values from I-RAVEN and use them to create the prompts for the model.

### New I-RAVEN-X

To further evaluate the mathematical reasoning capabilities at scale, we introduce an extension of the I-RAVEN's `center` constellation, called I-RAVEN-X. Our new benchmark maintains I-RAVEN's rules and attributes but allows for a parameterizable number of columns ($g$) and a dynamic range of attribute values ($m$). Appendix D provides details about the dataset generation, and Figure 2b shows example prompts from I-RAVEN-X.

## 3   LLM-based RPM solving

### Models

We focused our evaluations on text-only LLMs. There exist attempts (Mitchell, Palmarini, and Moskvichev 2024; Jiang et al. 2024; Cao et al. 2024; Ahrabian et al. 2024; Zhang et al. 2024) that leverage vision support of some multi-modal LLMs (e.g., GPT-4V) directly feeding the models with visual RPM data; however, they achieve consistently lower reasoning performance than with text-only prompting. The SOTA LLM-based abstract reasoning approach (Hu et al. 2023) relied on reading out GPT-3's (`text-davinci-002`) token probabilities. However, this model is no longer accessible to users, and its successive iterations did not allow prediction logits to be retrieved at the time of writing. Hence, we considered discrete classification approaches that are based on output strings rather than distribution over tokens. In particular, we investigated two SOTA LLMs: the proprietary GPT-4 (OpenAI et al. 2024)[2] (`gpt-4-0613`) and the open-source Llama-3 70B (Dubey

et al. 2024)[3]. More recent iterations of these models were not considered in our analysis for different reasons. Meta's attribution requirement in their updated terms regarding naming conventions prevented us from testing Llama-3.1 During initial tests, GPT-4o yielded worse results than GPT-4, hence we focused on GPT-4. Moreover, GPT-4 o1's poor abstract reasoning results on RPM (Latif et al. 2024) (18% on `2x2` RAVEN) and its limited availability (only preview version available at time of writing) prevented us from performing statistically significant tests on this chain-of-thought model.

### Prompting and classification

**Entangled and disentangled prompts**  Following (Hu et al. 2023), we evaluated two different prompting strategies, *entangled* and *disentangled* prompting. The entangled prompting provides all the attributes' values in a single prompt (see Appendix A). The disentangled prompting, on the other hand, is a compositionally structured approach that queries the LLM for individual attribute prediction. Disentangled prompting simplifies the task, but increases the number of queries by $3\times$.

**Discriminative and predictive classification**  Similarly to (Gendron et al. 2024), we consider two approaches to solve RPM tests with LLMs. In the *discriminative* approach, we provide the attribute descriptions of both the context matrix and the answer candidates. The LLM is then asked to return the panel number of the predicted answer. Appendix A provides an example prompt of the discriminative approach. In the *predictive* approach, we prompt the LLM only with the context matrix without the candidate answers. The LLM has to predict the value of the empty panel (see Figure 2). For selecting the final answer, we compare the predicted values with the answer panels and pick the one with the highest number of overlapping values. While the predictive approach may appear more difficult, it implicitly biases the

---

[2]GPT-4 was accessed between 07/03/2024–10/30/2024.

[3]The model weights were downloaded and evaluated locally.

LLM to approach the task as humans usually do, i.e., first applying a generative process to abduce rules and execute them to synthesize a possible solution, and then discriminatively selecting the most similar answer from choices (Holyoak and Morrison 2013). Moreover, the final answer selection is done without the intervention of the LLM, rendering phenomena like hallucinations less likely. Thus, the predictive classification can be seen as a more guided approach that helps LLM to solve the task.

**Additional enhancements**   Finally, we also employ well-known prompting-enhancing techniques such as self-consistency (Wang et al. 2023; Lewkowycz et al. 2022) and in-context learning (Brown et al. 2020) to improve the performances. More details are provided in Appendix A.

# 4   ARLC: learning abductive reasoning using VSA distributed representations

This section presents the Abductive Rule Learner with Context-awareness (ARLC), which performs neuro-symbolic reasoning with distributed VSA representations (see Figure 3). ARLC projects each panel's attribute value (or distributions of values) into a high-dimensional VSA space. The resulting VSA vectors preserve the semantic similarity between attribute values: the dot products between corresponding VSA encoded vectors define a similarity kernel (Plate 2003; Frady et al. 2022). Moreover, simple component-wise operations on these vectors, binding and unbinding, perform addition and subtraction respectively on the encoded values. For rule learning, ARLC introduces a generic rule template with several terms forming a series of binding and unbinding operations between vectors. The problem of learning the rules from data is reduced to a differentiable assignment problem between the terms of the general rule template and the VSA vectors encoding the contents of the panels, which can be learned with standard SGD. ARLC was initially presented in (Camposampiero et al. 2024); this work mainly compares it to the reasoning capabilities of LLMs on I-RAVEN, and demonstrates its extension to larger grid sizes and dynamic ranges on our novel I-RAVEN-X.

## From visual attributes to distributed VSA representations

ARLC's key concept is to represent attribute values with high-dimensional, distributed VSA vectors that preserve the semantic similarity between the attribute values thanks to an introduced notion of kernel. We start by defining a VSA that equips the space with dimensionality-preserving vector operations (binding $\otimes$, unbinding $\oslash$, and bundling $\oplus$) as well as a similarity function ($\text{sim}(\cdot, \cdot)$). For example, ARLC uses binary generalized sparse block codes (GSBCs) (Hersche et al. 2024b) as a particular VSA instance. In binary GSBCs, the $D$-dimensional vectors are divided into $B$ blocks of equal length, $L = D/B$, where only one (randomly selected) element per block is set to 1 ($D = 1024$ and $B = 4$). The algebraic operations of binary GSBCs are defined in Table 1. See Appendix B for a detailed background on VSA.

Next, we define a mapping $z : \mathbb{Z}^+ \to \mathbb{R}^D$ that enables the projection of input RPM attributes into a correspond-

ing high-dimensional, semantically-rich feature space. Note that this work focuses on mapping integer values as the attribute values in I-RAVEN are integer-valued too. However, generalizing this approach to real-valued domain mappings is possible using frequency holographic reduced representations (FHRR) (Plate 1995). Leveraging fractional power encoding (FPE) (Plate 2003), a value $v \in \mathbb{Z}^+$ is encoded as $\mathbf{z}(v) = \mathbf{z}^v = \bigotimes_{n=1}^{v} \mathbf{z}$, where $\mathbf{z} \in \mathbb{R}^D$ is a random binary GSBC vector. This mapping yields a similarity kernel between neighboring vector representations (Frady et al. 2022), as shown in Figure B.7 in Appendix B.

Let us assume two variables with values $v_1$ and $v_2$, which are represented with two VSA vectors ($\mathbf{z}(v_1) = \mathbf{z}^{v_1}$ and $\mathbf{z}(v_1) = \mathbf{z}^{v_2}$). Binding the two vectors yields $\mathbf{z}(v_1) \otimes \mathbf{z}(v_2) = \mathbf{z}^{v_1} \otimes \mathbf{z}^{v_2} = \mathbf{z}^{v_1 + v_2}$. Hence, binding in the VSA space is equivalent to the addition in $\mathbb{R}$. In other words, the FPE initialization allows to establish a semantic equivalence between high-dimensional vectors and real numbers. This property is consistently exploited in ARLC's framework, as it allows to solve the analogies in the RPM puzzles as simple algebraic operations in the domain of real numbers. For example, by computing the similarity between the bound representation and a third projected variable ($\text{sim}(\mathbf{z}^{v_1 + v_2}, \mathbf{z}^{v_3})$), we can evaluate if $v_1 + v_2 \overset{?}{=} v_3$ representing the `arithmetic plus` rule in RPM.

One advantage of performing reasoning with distributed VSA representations is its capability to represent perceptual uncertainty in the variable values. Connecting to the previous example, let us assume that the first variable takes value $v_1$ with probability $p$ and value $v_1'$ with probability $p' = 1 - p$. The distribution can be encoded as the weighted superposition of the two corresponding codewords: $p \cdot \mathbf{z}^{v_1} + p' \cdot \mathbf{z}^{v_1'}$. The similarity computation between the bound representation and a third variable would then yield

$$\text{sim}((p \cdot \mathbf{z}^{v_1} + p' \cdot \mathbf{z}^{v_1'}) \otimes \mathbf{z}^{v_2}, \mathbf{z}^{v_3}) =$$

$$p \cdot \text{sim}(\mathbf{z}^{v_1} \otimes \mathbf{z}^{v_2}, \mathbf{z}^{v_3}) + p' \cdot \text{sim}(\mathbf{z}^{v_1'} \otimes \mathbf{z}^{v_2}, \mathbf{z}^{v_3}),$$

where we use the linearity of the binding operation and the similarity metric. This formulation allows the validation of multiple solutions (in this case two) using only a single binding and similarity computation.

In the RPM application, each panel's label is translated to a probability mass function (PMF) $\mathbf{p}_a^{(i,j)}$, where $a$ is the attribute, $i$ is the row index and $j$ is the column index of the panel. The panel's PMF is then projected into the VSA space

$$\mathbf{v}_a^{(i,j)} = \sum_{k=1}^{m} \mathbf{p}_a^{(i,j)}[k] \cdot \mathbf{z}^k,$$

where $m$ is the number of possible values that the attribute $a$ can assume. Overall, this yields eight VSA vectors for each attribute $a$ (one for each panel of the input RPM matrix). Note that the basis vectors are pre-computed and stored in a dictionary $\mathbb{C} = \{\mathbf{z}^k\}_{i=1}^{r}$ containing $m$ elements.

## Learning RPM rules as an assignment problem

As we have seen in the previous example, RPM rules can be represented using VSA operations. Generalizing the application beyond the `arithmetic plus` rule, we find that

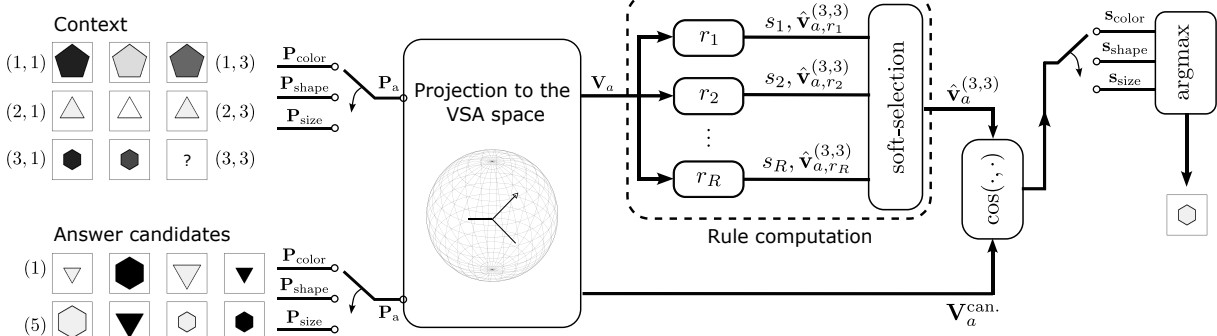

Figure 3: ARLC architecture. ARLC maps attribute values, or distributions of values, to distributed VSA representations, where the semantic similarity between values is preserved via a notion of kernel. Learnable rules ($r_1, ..., r_R$) predict the VSA representation of the empty panel ($\hat{\mathbf{v}}_{a,r}^{(3,3)}$) together with a confidence value ($s_r$). The closest answer to the predicted soft-selected prediction ($\hat{\mathbf{v}}_a^{(3,3)}$) is chosen as the final answer.

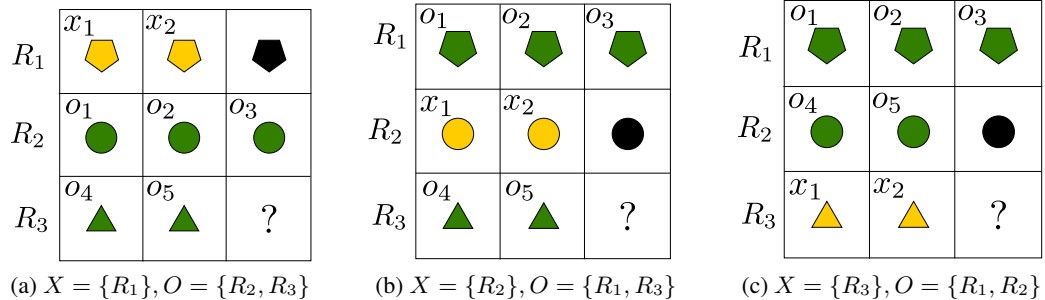

(a) $X = \{R_1\}, O = \{R_2, R_3\}$     (b) $X = \{R_2\}, O = \{R_1, R_3\}$     (c) $X = \{R_3\}, O = \{R_1, R_2\}$

Figure 4: Visualization of current samples ($X = \{x_1, x_2\}$, in yellow) and context ($O = \{o_1, \ldots, o_5\}$, in green) panels when predicting the third panel for different rows, namely the first row (left), second row (center) and third row (right). Black objects represent panels that are not used for the computation, while the question mark represents the unknown test panel, which is unavailable during inference.

Table 1: VSA operations and their equivalent in $\mathbb{R}$.

| Operation | Binary GSBCs with FPE | Equivalent in $\mathbb{R}$ |
|---|---|---|
| Binding ($\otimes$) | Block-wise circular convolution | Addition $+$ |
| Unbinding ($\oslash$) | Block-wise circular correlation | Subtraction $-$ |
| Bundling ($\oplus$) | Sum & normalization | — |
| Similarity ($\odot$) | Cosine similarity ($\cos(\cdot, \cdot)$) | — |

the rules used in RPM can be framed as a series of binding and unbinding operations:

$$r = (\mathbf{c}_1 \otimes \mathbf{c}_2 \otimes \mathbf{c}_3 \otimes \mathbf{c}_4 \otimes \mathbf{c}_5 \otimes \mathbf{c}_6) \oslash$$
$$(\mathbf{c}_7 \otimes \mathbf{c}_8 \otimes \mathbf{c}_9 \otimes \mathbf{c}_{10} \otimes \mathbf{c}_{11} \otimes \mathbf{c}_{12}), \quad (1)$$

where $\mathbf{c}_i$ can be assigned to a context panel $\mathbf{v}_a^{(i,j)}$ or the identity $\mathbf{e}$. In this setting, learning the rules of RPM can hence be interpreted as an assignment problem between VSA vectors and terms of Equation (1).

Motivated by works in cognitive sciences and psychology that argue for the importance of context in the solution of analogies for humans (Chalmers, French, and Hofstadter 1992; Cheng 1990), ARLC uses a general formulation of the soft-assignment problem which relies on the notion of

*context*:

$$\mathbf{c}_k = \sum_{i=1}^{I} w_k^i \cdot \mathbf{x}_i + \sum_{j=1}^{J} u_k^j \cdot \mathbf{o}_j + v_k \cdot \mathbf{e}, \quad (2)$$

where $\mathbf{w}, \mathbf{u}, \mathbf{v}$ are the learned parameters and they are subject to the following constraints:

$$\sum_{i=0}^{I} w_k^i + \sum_{j=0}^{J} w_k^j + v_k = 1,$$

$$0 \le w_k^i \le 1 \, \forall i, \quad 0 \le u_k^j \le 1 \, \forall j, \quad 0 \le v_k \le 1, \, \forall k.$$

Here, $\mathbf{X} = \{\mathbf{x}_1, \ldots, \mathbf{x}_I\}$ is the set of attributes that define the current sample, that is, the description of the problem for which we infer a solution. $\mathbf{O} = \{\mathbf{o}_1, \ldots, \mathbf{o}_J\}$ is the set of attributes that define the context for that sample, that could be interpreted as a working memory from which additional information to infer the answer can be retrieved. For predicting the empty panel in the last row, the context ($\mathbf{O}$) corresponds to the first two rows and the current samples ($\mathbf{X}$) to the last row (see Figure 4c). We augment this standard prediction with two more permutations, which aim to predict the last panel of the first and second row (see Figure 4a and Figure 4b). The knowledge of the right-most panels in the first

two rows allows us to compute a rule confidence by comparing the rule's prediction with the actual panel representation via the cosine similarity.

## Executing and selecting the learned rules

Inference with the learned rule set is a two-step process: an execution step (where all the rules are applied in parallel to the input) and a selection step (where a prediction for the missing panel is generated). The application of each rule $r$ to an RPM example generates a tuple of three VSA vectors $(\hat{\mathbf{v}}_{a,r}^{(i,3)})_{i=1}^3$, which corresponds to the result of the rule execution on the three rows of the RPM matrix, together with a rule confidence value $s_r$. The confidence value is computed as the sum of the cosine similarities between the predicted VSA vectors and their respective ground-truth vector,

$$s_r = \sum_{i=1}^{3} \cos\left(\mathbf{v}_a^{(i,3)}, \hat{\mathbf{v}}_{a,r}^{(i,3)}\right). \qquad (3)$$

During inference, the last term of the sum ($i = 3$) is omitted, as the ground truth for the third row is unknown.

The answer is finally produced by taking a linear combination of the VSA vectors generated by executing all the rules, weighted by their respective confidence scores (normalized to a valid probability distribution using a softmax function). More formally, if we define $\mathbf{s} = [s_1, \ldots, s_R]$ to be the concatenation of all rules' confidence score and $\hat{\mathbf{V}}_a^{(3,3)} = [\hat{\mathbf{v}}_{a,1}^{(3,3)}, \ldots, \hat{\mathbf{v}}_{a,R}^{(3,3)}]$ to be the concatenation of all rules' predictions for the missing panel, the final VSA vector predicted by the model for the attribute $a$ becomes

$$\hat{\mathbf{v}}_a^{(3,3)} = \text{softmax}\,(\mathbf{s}) \cdot \hat{\mathbf{V}}_a^{(3,3)}. \qquad (4)$$

The use of the weighted combination can be understood as a *soft-selection* mechanism between rules and was found to be more effective compared to the *hard-selection* mechanism provided by sampling (Hersche et al. 2024a).

## Training Loss and other Implementation Aspects

We follow the training recipe provided by Learn-VRF (Hersche et al. 2024a). The model is trained using stochastic gradient descent (SGD) with a learning rate lr = 0.01 for 25 epochs. The training loss is defined as the inverse cosine similarity between the three predicted panels and their corresponding ground truth

$$\mathcal{L} = 1 - \sum_{i=1}^{3} \cos\left(\mathbf{v}_a^{(i,3)}, \hat{\mathbf{v}}_a^{(i,3)}\right). \qquad (5)$$

As in Learn-VRF, we set the number of rules to $R = 5$. A single set of rules is instantiated and shared between all RPM attributes.

## Applying ARLC on I-RAVEN-X

While ARLC was initially designed for I-RAVEN, it can be seamlessly extended to our I-RAVEN-X with minor modifications. First, the number of binding/unbinding terms in Equation (1) is increased, e.g., from 12 to 22 to support the larger grid size of $g = 10$. Moreover, we increase the number of entries in the dictionary ($\mathbb{C}$) to support the larger dynamic range ($m$). Notably, only varying the dynamic range

Table 2: Task accuracy (%) on the `center` constellation of I-RAVEN. Among the baselines, we replicate Learn-VRF (Hersche et al. 2024a); the other results are taken from (Hersche et al. 2023). The standard deviations are reported over 10 random seeds. Llama-3 and GPT-4 are queried with the corresponding best prompting technique (see Table 3). Number of parameters for GPT-4 is not publicly available. The reasoning backend of PrAE, NVSA, and our ARLC$_{\text{p}\mapsto\text{l}}$ do not have trainable parameters.

| Method | Parameters | Accuracy |
|---|---|---|
| MLP (Hersche et al. 2024a) | 300 k | 97.6 |
| SCL (Wu et al. 2020) | 961 k | $99.9^{\pm 0.0}$ |
| PrAE (Zhang et al. 2021) | n.a. | $83.8^{\pm 3.4}$ |
| NVSA (Hersche et al. 2023) | n.a. | $99.8^{\pm 0.2}$ |
| Learn-VRF (Hersche et al. 2024a) | 20 k | $97.7^{\pm 4.1}$ |
| GPT-3 (Hu et al. 2023) | 175 b | 86.4 |
| Llama-3 | 70 b | 85.0 |
| GPT-4 | unk. | 93.2 |
| ARLC$_{\text{progr}}$ | n.a. | $100.0^{\pm 0.0}$ |
| ARLC$_{\text{learn}}$ | 480 | $98.4^{\pm 1.5}$ |

at constant grid size does not require retraining: we can simply replace the dictionary in order to support OOD generalization. Indeed, we could demonstrate that ARLC trained on a dynamic range of $m = 45$ can favorably generalize to a dynamic range of $m = 1000$.

# 5  Results

## Main results on I-RAVEN

Table 2 compares our LLM results with ARLC on the `center` constellation of I-RAVEN, also considering a range of neuro-symbolic and connectionist baselines. For the LLMs, we show the results with the corresponding best prompting techniques (see the ablation in the next subsection).

Moreover, we present results for two different versions of ARLC: ARLC$_{\text{progr}}$, where the model's weights are manually programmed with RPM rules ($R = 4$, since `constant` can be considered as a special case of `progression`), and ARLC$_{\text{learn}}$, where the rules are learned from scratch from data.

Among the LLM approaches, our GPT-4-based approach achieved the highest accuracy (93.2%) notably outperforming previous SOTA LLM-based abstract reasoning approaches on this benchmark (86.4%) (Hu et al. 2023). Yet, all LLM approaches fall behind the tailored connectionist and neuro-symbolic solutions. Notably, with only 480 learnable parameters, ARLC achieves a high accuracy of 98.4%.

## Ablation of LLM prompting techniques

Table 3 shows the task accuracy on I-RAVEN using GPT-4 and Llama-3 70B in various prompting configurations. Overall, both models benefit from the additional guidance provided by our prompting techniques. Concretely, using a predictive approach and querying for individual disentangled attributes yielded already high accuracies (91.4% and

Table 3: Task accuracy (%) on the `center` constellation of I-RAVEN ablating various LLM prompting techniques.

| Predictive/ discriminative | Disentangled queries per attribute (3×queries) | Self-consistency (n=7) | In-context learning (s=16) | GPT-4 | Llama-3 70B |
|---|---|---|---|---|---|
| Discriminative | | | | 56.0 | 22.8 |
| Discriminative | ✓ | | | 60.0 | 22.4 |
| Predictive | | | | 74.8 | 79.0 |
| Predictive | ✓ | | | 91.4 | 83.2 |
| Predictive | ✓ | ✓ | | **93.2** | 84.8 |
| Predictive | ✓ | | ✓ | 85.4 | 84.8 |
| Predictive | ✓ | ✓ | ✓ | 86.4 | **85.0** |

Table 4: Accuracy (%) of predicting the correct attribute value. Results are averaged across attributes.

| Model | Disentangled queries per attribute (3×queries) | Constant | Progression | Distribute three | Arithmetic |
|---|---|---|---|---|---|
| GPT-4 | No | 100 | 98.0 | 91.6 | 27.1 |
| | Yes | 100 | 100 | 99.5 | 73.6 |
| Llama-3 70B | No | 100 | 97.2 | 99.3 | 31.0 |
| | Yes | 100 | 100 | 96.6 | 45.0 |

83.2% for GPT-4 and Llama-3 70B, respectively). Introducing self-consistency further improves the accuracy for both models. Llama-3 70B's performance can be further pushed (to 85.0%) by using self-consistency and in-context learning. On the contrary, GPT-4 cannot make use of the additional in-context samples, yielding a lower accuracy instead.

**LLMs show weakness in arithmetic rule**

Even though both LLMs achieve a reasonable overall task accuracy, they fail in some instances. We shed more light on the reasoning capability of the two models by analyzing the accuracy of predicting the correct value for a given rule. As shown in Table 4, both models perform well on `constant`, `progression`, and `distribute three` rules, whereas the accuracy notably drops for the `arithmetic` rule. One explanation for the accuracy drop could be the LLM's tendency for (short-sighted) relational reasoning, instead of performing relational mapping that requires the understanding of the first two rows before applying a rule on the last row (Stevenson et al. 2023). We analyze this hypothesis in Appendix C, where we attempt to explain the LLM's wrong predictions by rules that may have been inferred from the last row. For GPT-4, 32 out of 68 errors can be explained by rules that might have been inferred from a partial context matrix, e.g., a `constant` or `progression` rule based on the last row.

**Results on our novel I-RAVEN-X**

Finally, we conduct experiments on our novel I-RAVEN-X test, which allows us to configure the matrix size and the dynamic range of the attribute values. We fix the grid size to $3 \times 10$ and vary the dynamic range between 50, 100, and 1000. As shown in Table 5, the performance on the `arithmetic` rule drops not only due to the larger grid size but also generally degrades with an increasing dynamic range: the arithmetic accuracy falls below 10% for

Table 5: Arithmetic accuracy (%) on I-RAVEN and our novel I-RAVEN-X. The LLMs use self-consistency (n=7). For ARLC$_{learn}$ we report max/mean evaluation accuracies over 5 different training seeds.

| | I-RAVEN $3 \times 3$ | I-RAVEN-X $3 \times 10$ | | |
|---|---|---|---|---|
| Dynamic range | 5–10 | 50 | 100 | 1000 |
| Llama-3 70B | 45.0 | 1.5 | 2.6 | 0.4 |
| GPT-4 | 73.6 | 30.4 | 25.1 | 8.4 |
| ARLC$_{progr}$ | 100.0 | 99.8 | 100.0 | 99.5 |
| ARLC$_{learn}$ | 99.5/99.2 | 99.1/95.5 | 98.9/96.3 | 97.9/95.3 |

both LLMs at the highest dynamic range (1000). At the same time, our ARLC maintains a high accuracy across the board, while only being trained at dynamic range of 50 and reconfigured for the higher ranges. Appendix D shows that the same trend holds for the overall task accuracy.

## 6  Conclusion

This work revealed LLM's limitations in recognizing and executing arithmetic rules in abstract reasoning tasks, despite being provided disentangled prompts with ground-truth visual attributes and using advanced prompting techniques. We further showed the serious limitation on a larger ($3 \times 10$) RPM test. As a viable alternative, we presented a neuro-symbolic approach (ARLC) that achieves a high accuracy both on I-RAVEN and our I-RAVEN-X, thanks to learning to reason with distributed VSA representations and operators. We hope that our findings will lead to the development of architectures that aim to improve reasoning capabilities, e.g., by integrating symbolic solvers such as our ARLC into LLMs.

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

# A Prompting details

This appendix provides more details on our prompting strategy. While the prompt design was mainly inspired by (Hu et al. 2023), we extended it with predictive and discriminative classification and fine-tuned it for the different models. For example, we found that adding a prefix ("Only return the missing number") helped to slightly improve GPT4's accuracy, whereas it reduced Llama-3 70B's performance. Thus, we used individual prompts for the different models.

## Joint attribute querying

As an alternative to individually querying the LLM for predicting the separate attributes, we also devised a joint attribute prompting scheme, shown in Figure A.5. The attributes of each panel are represented in brackets: (shape, size, color). In this setting, the LLM is required to predict all three attributes of the missing panel at once. For better distinguishing between the different attributes, they are scaled with individual factors ($1\times$, $0.1\times$, $10\times$).

## Discriminative classification approach

Figure A.6 shows an example prompt for performing discriminative classification. As shown, the answers only contain two distinct values ("6" and "7"); finding the correct answer requires the consideration of all attributes. For choosing the final answer, we extract all attribute values that correspond to the predicted answer (e.g., value "7" for shape) and select the best matching answer candidate, i.e., the answer with the highest number of overlaps with the predicted attributes.

## Self-consistency

As an optional extension, we employ self-consistency (Wang et al. 2023; Lewkowycz et al. 2022) by querying the model multiple times ($n = 7$ times), sampling the next token from the distribution with a non-zero soft-max temperature. We find the optimal soft-max temperature for GPT-4 ($T = 0.5$) and Llama-3 70 B ($T = 0.4$) via a grid search on a subset of 50 I-RAVEN problems. We did not explore the effect of other parameters, such as top-k or top-p, and set them to the default values. The final prediction is determined by a majority vote over the sampled outputs. The selection of an odd number of samples (i.e., $n = 7$) helps to prevent potential ties.

## In-context learning

For a better understanding of the RPM task, we optionally prefix 16 in-context examples to the prompt (Brown et al. 2020). In the predictive classification approach (where no answer candidates are provided), we simply provide complete example RPM matrices. The in-context samples are randomly selected from I-RAVEN's training set. Examples that had the same context matrix as the actual task are discarded and re-sampled to prevent shortcut solutions.

# B Vector-symbolic architectures

Vector-symbolic architectures (VSAs) (Plate 1995, 2003; Gayler 2003; Kanerva 2009) are a family of computational models that rely on the mathematical properties of high-dimensional vector spaces. VSAs make use of high-dimensional distributed representations for structured (symbolic) representation of data while maintaining the advantages of connectionist distributed vector representations (see (Kleyko et al. 2023) for a survey). Here is a formal definition of VSAs:

**Definition 1** (VSA). *A vector-symbolic architecture (VSA) consists of a 4-tuple $\mathbb{V} = (\mathbb{C}, \oplus, \otimes, \odot)$, where $\mathbb{C}$ is a set of high-dimensional distributed vectors equipped with two main operations, $\oplus$ (bundling) and $\otimes$ (binding), and on which it is possible to define a similarity measure $\odot$.*

Bundling is a similarity-preserving operation that creates a superposition of the operands, that is, the resulting vector will have a high similarity with the two operands. Binding, on the other hand, is an operation that allows to bind a vector (value) to another vector (key) and does not preserve similarities; it usually allows an inverse operation, called unbinding. The specific realization of the bundling, binding, and vector space constitute the main difference between members of the VSA family.

# C Analysis of arithmetic errors

This appendix aims to find explanations for LLM's errors by analyzing the structure behind the predicted answers. A recent study (Stevenson et al. 2023) showed that LLMs tend to solve verbal analogy problems in an associative way instead of performing proper relational mapping. The associative reasoning can be explained as ignoring the source domain and solving the task directly at the target domain (e.g., only looking at the possible solutions without reading the questions). Interestingly, children tend to perform associative reasoning, whereas adults opt for relational mapping.

In RPMs, the source domain can be defined as the first two rows (with values $x_{1,1}, x_{1,2}, x_{1,3}$ and $x_{2,1}, x_{2,2}, x_{2,3}$), whereby the target domain is the last row ($x_{3,1}, x_{3,2}$). Therefore, an associative reasoner would only look at the last row to solve the task. In the following, we aim to find potential incorrect rules that the LLMs may have been inferred from the last row(s):

- constant: The values of the last row are identical ($x_{3,1} = x_{3,2}$), and the model predicts $\hat{x}_{3,3} = x_{3,2} = x_{3,1}$
- progression: The values of the last row differ by $\delta = x_{3,2} - x_{3,1}$, and the model predicts $\hat{x}_{3,3} = x_{3,2} + \delta$
- short constant: The model just copies the penultimate value: $\hat{x}_{3,3} = x_{3,2}$.
- short distribute three: Assuming a distribute three over the last two rows: $x_{3,1} \in \{x_{2,1}, x_{2,2}, x_{2,3}\}$, $x_{3,2} \in \{x_{2,1}, x_{2,2}, x_{2,3}\}$, and hence $\hat{x}_{3,3} \in \{x_{2,1}, x_{2,2}, x_{2,3}\}$.

Figure C.8 shows the resulting confusion matrix summarizing all the attributes. The arithmetic rule has fewer occurrences as this rule is not integrated in the attribute shape. As already stated in the main text, the majority of wrong predictions are related to the arithmetic rule. For GPT-4, our new rule interpretations can explain 32 out of the 68 errors, while 36 errors remain unknown. Llama-3 70B showed many more errors in the arithmetic rule; here, we can explain 57 out of 142 errors with relational reasoning.

```
System: Complete the Raven's progressive matrix:
User:    Only return the missing numbers!
         row 1: (3,0.5,50), (6,0.5,50), (4,0.5,50);
         row 2: (4,0.3,10), (3,0.3,10), (6,0.3,10);
         row 3: (6,0.1,70), (4,0.1,70), (
Out:     3,0.1,70)
```

Figure A.5: Example prompt for joint prediction of all three attributes.

```
System: Complete the Raven's progressive matrix:
User:    row 1: 4, 4, 4;
         row 2: 6, 6, 6;
         row 3: 7, 7,

         Select the correct Answer from the following list
         Answer #0: 7
         Answer #1: 6
         Answer #2: 7
         Answer #3: 7
         Answer #4: 6
         Answer #5: 6
         Answer #6: 7
         Answer #7: 6

         Solution: The correct answer is Answer #
Output:  0: 7
```

Figure A.6: Example prompt for discriminative classification approach, where the answer candidates are provided. The underlying attribute is `shape` and the rule is `constant`.

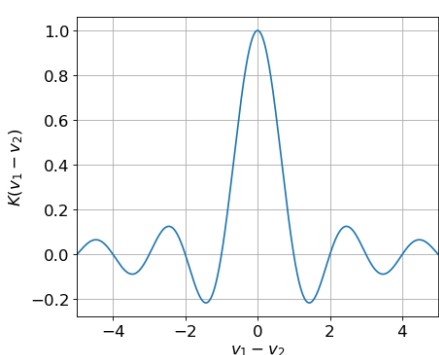

Figure B.7: Similarity kernel in VSA. Mapping two values ($v_1$ and $v_2$) to a VSA space (i.e., GSBC in ARLC) that uses fractional power encoding (FPE) and computing their similarity in the VSA space yields the shown similarity kernel $K(v_1 - v_2)$.

In summary, some (40.1–47.1%) of the LLM's errors can be rooted in relational reasoning. Further understanding the behavior of the unknown rules is scope for future work.

## D   I-RAVEN-X

This appendix provides more details on the generation of I-RAVEN-X, as well as more results on the task accuracy in Table D.6.

When generating a new RPM example, we uniformly sample from one of the available rules (`constant`, `progression`, `arithmetic`, and `distribute three`). Note that the attribute `shape` does not incur the `arithmetic` rule. We use I-RAVEN's attribute bisection tree (Hu et al. 2021) to generate unbiased candidate answers. In the following, we describe the context matrix generation for the individual rules of our new I-RAVEN-X dataset. The overall goal is that the values stay in the range $[0, m - 1]$.

- `constant`: This rule keeps the attribute value constant per row. For each row, we uniformly sample an integer from the set $\{0, 1, ..., m - 1\}$, and duplicate along the row.

- `progression`: The attribute value monotonically increases or decreases in a row by a value of 1 or 2. First, we uniformly sample the progressive increment/decrement ($\delta$) from the set $\{-2, -1, +1, +2\}$. In case of a positive increment, we first define the values of the right-most columns, by uniformly sampling from the set

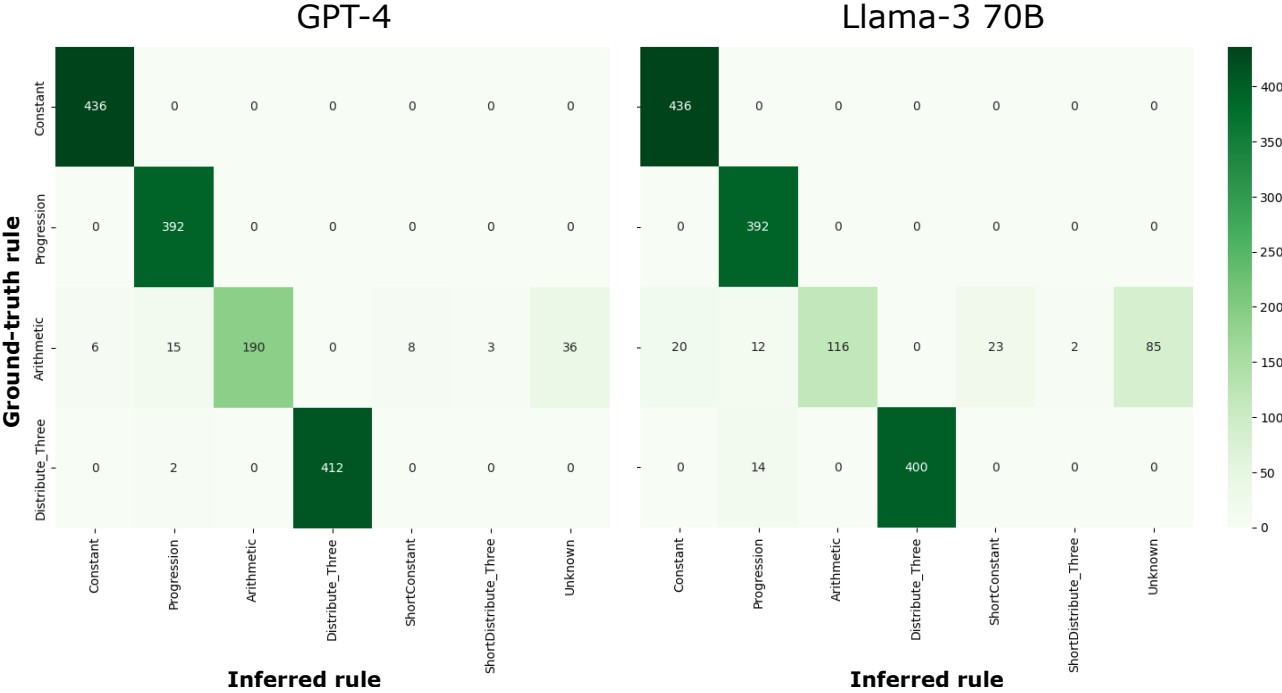

Figure C.8: Rule confusion matrix of GPT-4 (left) and Llama-3 70B (right).

$\{(g-1)\cdot\delta, ..., m-1\}$ for each row. Then, the rest of the matrix is completed by applying the progression rule. The sampling for a negative $\delta$ is done specularly from the first column.

- arithmetic: The attribute values of the first $g-1$ panels are either added (arithmetic plus) or subtracted (arithmetic minus), yielding the attribute value of the last panel in the row. In arithmetic plus, we sequentially sample the values from the first $g-1$ panels in the row. For each panel, we set the sampling range to $\{0, ..., m-s\}$, where $s$ is the sum of the already sampled panels in the row. Afterward, the first $g-1$ panels are shuffled. Finally, the values of the last panels are the sum of the first $g-1$ ones, applied row-wise. For arithmetic minus, we apply the same sampling strategy but leave the first column empty. The value of the first column is then defined as the sum of the other columns.

- distribute-n: We uniformly sample distinct values for the first row from $\{0, ..., m-1\}$. The content of the remaining rows is defined by applying a circular shift per row (either right or left).

Table D.6: Task accuracy (%) on I-RAVEN and our novel I-RAVEN-X. The LLMs use self-consistency (n=7). For ARLC$_{\text{learn}}$ we report max/mean evaluation accuracies over 5 different training seeds.

| | I-RAVEN $3 \times 3$ | I-RAVEN-X $3 \times 10$ | | |
|---|---|---|---|---|
| Dynamic range | 5–10 | 50 | 100 | 1000 |
| Llama-3 70B | 85.0 | 76.8 | 73.0 | 74.2 |
| GPT-4 | 93.2 | 82.2 | 79.6 | 76.6 |
| ARLC$_{\text{progr}}$ | 100.0 | 100.0 | 100.0 | 99.7 |
| ARLC$_{\text{learn}}$ | 99.1/98.6 | 94.6/86.3 | 95.1/88.0 | 91.6/82.8 |