# OpenReview forum: "Towards Learning to Reason: Comparing LLMs with Neuro-Symbolic on Arithmetic Relations in Abstract Reasoning"
_AAAI.org/2025/Workshop/NeurMAD — AAAI 2025 Workshop NeurMAD Submission_

### Official Review · Reviewer_5UzQ · 2024-12-20
**The paper compares the performance of large language models (LLM), specifically GPT-4 and Llama 3, with a neuro-symbolic approach that combines vector symbolic architecture and abductive rule learning, focusing on the task of abstract reasoning using the Raven dataset.**

**Rating:** 6
**Confidence:** 5

**Review:**

The paper is interesting as it explores the application of vector symbolic architecture and abductive rule learning in abstract reasoning tasks. It would be better if the following points can be clarified:

(1) Equations 2 and 3 are confusing with respect to the variable names used. For instance, what is c1, c2..c6,..c12? It is mentioned that c_i represents v_a at (i,j), so shouldn't it be cij? I suppose i and j represents row and column numbers respectively.

Equation 3 is also not clear - what is I and j - is it number of rows and columns again?

(2)  The three variants of ARLC - ARLC_progr, ARLC_learn and ARLC_p->1, ARLC_progr has some knowledge about the rules, and ARLC_p->1 which is initialised with programmed rules shows lower accuracy than ARLC_learn which learns all rules from scratch (see Table 2) - how do you explain this? Why doesn't Tables 5 and 6 show results on ARLC_p->1? It would help if you could give more details on manual programming of weights and rule initialisation to better understand the different variants.

(3) While comparing ARLC_progr and ARLC_p->1 with LLM, isn't it fairer if we provide some knowledge about the rules to LLM as well? How would the LLM perform if it has access to the rules?

(4)Can we apply other interpretable rule learning frameworks like ProbFOIL (Inducing Probabilistic Relational Rules from Probabilistic Examples, Luc de React et al) for the task? What is the significance of VSA? It would be easier to understand if the paper could also show examples of the rules learned by the approach.

---

### Official Review · Reviewer_MjJi · 2024-12-26
**Nice comparison**

**Rating:** 6
**Confidence:** 2

**Review:**

This paper describes a comparison between leading LLMs (GPT-4 and Llama-3) and the Abductive Rule Learner with
Context-awareness (ARLC), a neuro-symbolic approach, in solving Raven’s progressive
matrices (RPM). In particular, the authors extend the RPM tests to larger matrices, so that out-of-distribution evaluation can be carried to better test the models reasoning ability.

Strength:
- comprehensive empirical study of various models on this interesting reasoning task.
- mostly well-written, not hard to follow.

Weakness:
- not much to take away as LLMs are already known to be bad at arithmetic reasoning
- it would be nice to see a few more words about why the I-RAVEN dataset matters and if the ARLC can be applied beyond this RPM task.

---

### Decision · Program_Chairs · 2024-12-30

**Decision:**

Accept

**Comment:**

This paper compares LLMs and neuro-symbolic methods for RPM tasks and may pave the way towards novel pure neuro-symbolic unification methods for visual logical reasoning.